# OpenReview forum: "LLM Unlearning with LLM Beliefs"
_ICLR.cc/2026/Conference — ICLR 2026 Poster_

### Official Review · Reviewer_8qgu · 2025-10-27

**Soundness:** 3
**Presentation:** 3
**Contribution:** 3
**Rating:** 6
**Confidence:** 3

**Summary:**

This paper investigates the "spurious unlearning" problem that occurs when LLMs perform unlearning tasks. The authors point out that while existing methods (such as gradient ascent and NPO) can reduce the probability of a target response, they also redistribute the probability mass toward semantically similar regions, resulting in a "squeezing effect."
To address this issue, the authors propose a Bootstrapping (BS) framework that incorporates the model's own high-confidence generation (model belief) into the forgetting objective.
BS-T suppresses high-probability tokens at the word level;
BS-S suppresses high-confidence generation of entire segments at the sequence level.
The authors validate their approach on benchmarks such as TOFU, MUSE, and WMDP, and provide a theoretical analysis of BS-T to explain its mechanism for mitigating the squeezing effect.

**Strengths:**

- Interesting and Important Discovery: The paper reveals the "squeezing effect" and the resulting "spurious unlearning", suggesting that current unlearning methods only achieve superficial unlearning and further analyzing the reasons.

- Simple Yet Effective Design: The Bootstrapping framework directly targets areas of high probability of false forgetting by jointly suppressing the target response and the model's own high-confidence output. It requires no additional models or external data and is logically self-consistent.

- Comprehensive Experimental Results: Systematic experiments on multiple datasets and models, compared with strong baselines such as NPO, WGA, and RMU, show consistent improvement. Qualitative examples and gradient dynamics analysis are also provided to further strengthen the demonstration.

**Weaknesses:**

- BS-S has high computational overhead: Sequence-level bootstrapping requires generating multiple belief sequences for each sample, significantly increasing computational costs. The paper does not provide clear time or resource costs, nor does it discuss scalability for large-scale applications.

- BS-S lacks theoretical support: The authors provide a gradient analysis for BS-T, but BS-S is validated solely by experimental results and lacks formal explanations or convergence guarantees.

- High-confidence suppression strategies carry risks: High confidence does not necessarily indicate content that should be forgotten. The top-k outputs of the model may contain semantically related but harmless tokens; ranking by probability alone may lead to excessive forgetting.

**Questions:**

This paper identifies and addresses flaws in existing LLM unlearning methods, using a sound approach and robust results.

To further enhance the paper's contributions, I have the following comments:

1. Quantitatively demonstrate the effectiveness of mitigating the "squeezing effect," such as the change in semantic similarity between generated samples before and after forgetting;

2. Report the computational overhead of BS-T and BS-S, and discuss their applicability in large-scale scenarios.

3. Supplement sensitivity and ablation analysis of the hyperparameters λ, k, and N;

4. Clarify the model belief sampling strategy and evaluate the impact of different parameter settings on the results;

5. Explore dynamic k or entropy-based adaptive strategies to mitigate the over- or under-forgetting issues associated with a fixed k.

---

> ### Author Response · Authors · 2025-11-24
> **Response to Reviewer 8qgu (Part 1/5)**
>
> Thank you for highlighting the importance of the squeezing effect, the simplicity of the bootstrapping design, and the comprehensiveness of our experiments. We address each of your weaknesses and questions point-by-point below.
>
> ---
> ### **Weakness 1: BS-S has high computational overhead**
> > *“BS-S has high computational overhead: Sequence-level bootstrapping requires generating multiple belief sequences for each sample, significantly increasing computational costs. The paper does not provide clear time or resource costs, nor does it discuss scalability for large-scale applications.”*
>
> Thank you for the comment. The paper **already reports concrete runtime measurements in Appx. F.6 (Fig. 6)**, and these show that BS-S remains comparable to standard baselines and substantially cheaper than heavier methods. For large-scale applications, we also outline practical strategies such as **offline belief pre-computation** and using **small** $N$ that keep BS-S scalable in real deployments.
>
> 1. **Clarifying the actual computational overhead**
>
>    Appx. F.6 (Fig. 6) reports per-epoch training time on TOFU 10% with LLaMA-3.2-1B-Instruct. For clarity, we present the values in **Table D.1**.
>
>    **Table D.1: Training time per epoch (Fig. 6).**
>    Method|Time (s)
>    -|-:
>    GradDiff|39
>    NPO|73
>    RMU|30
>    WGA|34
>    SimNPO|54
>    UNDIAL|36
>    GRU|225
>    BS-T|33
>    BS-S|64
>
>    **Table D.1** shows that:
>    * **BS-S (64s)** is *faster than NPO (73s)* and dramatically faster than heavy approaches such as GRU (225s).
>    * Although slower than the lightest baselines (e.g., RMU/WGA), the overhead remains **well within practical bounds**.
>
>    Also, since LLM unlearning itself is a much more efficient technique compared to alignment, the computational overhead of our method is practical and acceptable.
>
> 2. **Scalability to large-scale applications**
>
>    To make BS-S efficient for large deployments, we recommend two practical strategies:
>    * **Offline belief pre-computation.** Run a single forward pass over the forget prompts to cache the $N$ high-confidencecontinuations. Training can then proceed **without any on-the-fly sampling**, removing most of the runtime overhead associated with BS-S.
>    * **Use small $N$.** As shown in Fig. 5(b), $N=1–3$ already captures the semantic variations needed for effective unlearning. Therefore, users can adjust $N$ according to the computational budget in practice.
>
>    With these strategies, BS-S scales cleanly to **large forget sets and large model sizes**, making it practical for real-world applications. We will include these discussions in the final version.

---

> ### Author Response · Authors · 2025-11-24
> **Response to Reviewer 8qgu (Part 2/5)**
>
> ### **Weakness 2: BS-S lacks theoretical support**
> > *“BS-S lacks theoretical support: The authors provide a gradient analysis for BS-T, but BS-S is validated solely by experimental results and lacks formal explanations or convergence guarantees.”*
>
> Thank you for raising this point. We have **updated the manuscript**, and clarified the theoretical standpoint of BS-S by **(1)** distinguishing off-policy and on-policy variants, **(2)** providing a learning-dynamics analysis for the off-policy case (new **Sec. 5.2**), and **(3)** explaining why the on-policy variant falls outside the assumptions of the AKG framework (**Appx. D.4**). We also clarify that **convergence guarantees are not meaningful for unlearning losses** that deliberately maximize NLL and thus inherently diverge. Below we summarize the key ideas; full analyses and proofs appear in Sec. 5 and Appx. D.
>
> 1. **Off- vs. on-policy BS-S**
>
>    In **Sec. 4.2** of the revised manuscript, we clarify that BS-S naturally divides into two regimes:
>    * **Off-policy BS-S:** high-confidence sequences are sampled *once* before finetuning and remain fixed.
>    * **On-policy BS-S:** sequences are re-sampled as model parameters update.
>
>    This distinction is crucial because the AKG learning-dynamics framework requires **teacher forcing** and a **fixed training set**. Under these conditions, off-policy BS-S becomes amenable to analysis, whereas on-policy BS-S does not.
>
> 2. **Theoretical analysis of off-policy BS-S**
>
>    In **Sec. 5.2** of the revised manuscript, we provide a theoretical perspective for **off-policy BS-S**. We show that BS-S corresponds to applying BS-T on an *expanded set of belief-aligned continuations*. Under the AKG framework, this induces an update that is a **kernel-weighted sum of BS-T residuals** across these sequences (**Thm. 5.3**). Intuitively:
>    * **BS-T** spreads forgetting across a *local* neighborhood defined by token-level beliefs.
>    * **Off-policy BS-S** aggregates these effects over *multiple high-likelihood continuations*, yielding **broader and smoother suppression** of sequence-level memorization.
>
>    This provides a principled justification for why sequence-level bootstrapping can outperform token-level suppression in practice.
>
> 3. **Discussion on on-policy BS-S**
>
>    For **on-policy BS-S**, we explicitly explain in **Appx. D.4** why its dynamics cannot be directly analyzed under AKG: the auxiliary sequences **depend on evolving parameters**, **breaking the teacher forcing assumption** required by the theory.
>
> 4. **On “convergence guarantees”**
>
>    Classical convergence guarantees do not apply to unlearning methods—GA, NPO-style losses, and BS variants all intentionally **maximize negative log-likelihood** on the forget set. Such objectives are **designed to diverge**, not converge, and unlearning is performed only for a few epochs in practice. Thus, expecting convergence guarantees is not appropriate for this class of objectives.
>
> Overall, the new revision adds the missing theoretical standpoint by introducing off-policy BS-S analysis and clarifying the limitations for the on-policy variant. We hope these additions address your concern and make the conceptual advantage of BS-S much clearer.

---

> ### Author Response · Authors · 2025-11-24
> **Response to Reviewer 8qgu (Part 3/5)**
>
> ### **Weakness 3: Risks of high-confidence suppression**
> > *“High-confidence suppression strategies carry risks: High confidence does not necessarily indicate content that should be forgotten. The top-$k$ outputs of the model may contain semantically related but harmless tokens; ranking by probability alone may lead to excessive forgetting.”*
>
> Thank you for the comment. We agree that probability-based selection may include some semantically related but harmless tokens. In practice, our design already mitigates this risk through **prompt-local bootstrapping, retain regularization, and conservative weighting**. We also outline optional extensions such as semantic filtering or reward-model scoring, which can further reduce excessive forgetting.
>
> 1. **Why excessive forgetting is limited in practice**
>
>    While top-$k$ tokens may contain benign items, our framework constrains their influence through three mechanisms:
>    * **Prompt-local bootstrapping.** Suppression is applied *only* in the specific context $x_u$. Even if a token is benign in general (e.g., occupations, attributes), it is penalized only *when it appears under the sensitive prompt*, which is aligned with the goal of removing the unwanted association rather than deleting global knowledge.
>    * **Retain regularization.** The retain loss provides a strong counter-gradient that preserves general knowledge and prevents broader degradation outside the forget domain. This is a key protection shared by strong unlearning baselines.
>    * **Conservative bootstrapping weights $\lambda$.** We intentionally use moderate $\lambda _ {\rm BST}$ and $\lambda _ {\rm BSS}$, keep suppression bounded.
>
>    These design choices ensure that high-confidence suppression remains localized, bounded, and utility-preserving, even if some benign tokens appear in the belief neighborhood.
>
> 2. **Future extensions to further reduce collateral suppression**
>
>    Our framework can naturally incorporate additional filtering to make BS-S even more selective:
>    * **LLM-based semantic vetting** of bootstrapped sequences before inclusion;
>    * **Reward-model scoring** to include only continuations truly aligned with the sensitive fact;
>    * **Embedding- or similarity-based thresholds** to exclude semantically distant or benign variants.
>
>    These design can further reduce the inclusion of harmless content. We will mention these as practical extensions in the final version.
>
> ---
> ### **Question 1: Quantitatively demonstrate mitigation of squeezing**
> > *“Quantitatively demonstrate the effectiveness of mitigating the ‘squeezing effect’, such as the change in semantic similarity between generated samples before and after forgetting.”*
>
> Thank you for the suggestion. We have **updated the manuscript**, and the revised manuscript now includes explicit quantitative evidence of squeezing mitigation through the **new Fig. 4**, which jointly analyzes **probability dynamics and semantic similarity**. These complementary results show that BS-T and BS-S suppress squeezing more effectively than existing baselines.
>
> 1. **Probability-level evidence (Fig. 4(a, b))**
>
>    **Fig. 4(a, b)** presents the log-probability dynamics for BS-T and BS-S using the same high/mid/low likelihood grouping as earlier figures. Both methods **monotonically decrease** the log-probabilities of the **target token** and its **high-likelihood paraphrase cluster**, demonstrating that probability mass is not redirected into semantically proximate regions. This provides quantitative evidence that the squeezing effect is mitigated rather than amplified.
>
> 2. **Semantic-level evidence (Fig. 4(c))**
>
>    **Fig. 4(c)** reports LaaJ Naturalness and Similarity scores, which directly measure semantic drift. BS-T and BS-S consistently obtain strong performance in LaaJ metrics.
>
> Together, these results demonstrate quantitative mitigation of both **(i)** likelihood-level squeezing and **(ii)** semantic-level spurious unlearning.
>
> ---
> ### **Question 2: Report computational overhead and large-scale applicability**
> > *“Report the computational overhead of BS-T and BS-S, and discuss their applicability in large-scale scenarios.”*
>
> See the response to **Weakness 1**.

---

> ### Author Response · Authors · 2025-11-24
> **Response to Reviewer 8qgu (Part 4/5)**
>
> ### **Question 3: Sensitivity and ablations on $λ$, $k$, and $N$**
> > *“Supplement sensitivity and ablation analysis of the hyperparameters $\lambda$, $k$, and $N$.”*
>
> Thank you for the suggestion. **Fig. 5 in Appx. F.5** already provides detailed ablations over $\lambda$, $k$, and $N$. To make the results easier to view on OpenReview, we convert Fig. 5(a, b) into **Tables D.2 and D.3** below. Overall, BS-T and BS-S exhibit **stable, high performance across broad hyperparameter ranges**, indicating that the methods are not overly sensitive and are straightforward to tune.
>
> 1. **BS-T sensitivity w.r.t. $\lambda$ and $k$**
>
>    **Table D.2: BS-T aggregate score across $\lambda$ and $k$ (Fig. 5(a)).**
>     $k$|$λ$ = 0.0|0.1|0.2|0.3|0.4|0.5
>     -:|-:|-:|-:|-:|-:|-:
>     5|0.50|0.52|**0.57**|0.46|0.34|0.31
>     10|0.50|**0.59**|**0.63**|0.46|0.35|0.33
>     20|0.50|0.56|**0.62**|0.43|0.41|0.37
>     30|0.50|0.51|0.56|0.43|0.34|0.34
>     40|0.50|0.51|0.57|0.42|0.39|0.34
>     50|0.50|0.50|0.56|0.46|0.33|0.32
>
>    **Table D.2** shows that BS-T performs consistently well for $\lambda \in [0.1, 0.3]$. This wide plateau shows the method is **robust**, with performance degrading only at extreme values.
>
> 2. **BS-S sensitivity w.r.t. $\lambda$ and $N$**
>
>    **Table D.3: BS-S aggregate score across $\lambda$ and $N$ (Fig. 5(b)).**
>     $N$|$λ$ = 0.0|0.2|0.4|0.5|0.6|0.8
>     -:|-:|-:|-:|-:|-:|-:
>     1|0.63|0.61|0.60|0.56|0.60|0.59
>     2|0.63|0.62|0.61|0.58|0.61|0.60
>     3|0.63|0.62|0.61|0.59|**0.63**|0.61
>     4|0.63|0.62|0.62|0.60|**0.64**|0.62
>     5|0.63|0.62|**0.63**|0.61|**0.64**|0.63
>     10|0.63|0.63|0.63|0.62|**0.64**|0.63
>
>    **Table D.3** shows that BS-S remains strong for $\lambda \in [0.2, 0.6]$ across all values of $N$. Improvements saturate for $N>3$, indicating that tuning is simple and the method is not sensitive to $N$.
>
> Overall, both BS-T and BS-S show **wide stable regions** in their hyperparameters, showing that the methods are robust and easy to tune in practice.
>
> ---
> ### **Question 4: Clarify model belief sampling strategy and impact of settings**
> > *“Clarify the model belief sampling strategy and evaluate the impact of different parameter settings on the results.”*
>
> Thank you for the question. We clarify below **(1)** how belief sequences are sampled in BS-S and **(2)** how different parameters influence unlearning performance. The sampling procedure is lightweight and stable, and the key sensitivities ($λ$ and $N$) are already quantified in Fig. 5(b).
>
> 1. **Sampling procedure in BS-S**
>
>    For each forget prompt $\mathbf{x} _ {\rm u}$, BS-S uses a simple and standard belief-sampling pipeline:
>    * **Generate $N$ sequences** using the current model with top-$k$, temperature-controlled decoding and a fixed maximum length.
>    * Feed the resulting auxiliary set $\hat{\mathcal{D}} _ {\rm u}$ into the **same unlearning loss** used for the original forget targets.
>    * **Resample once per epoch**, although this rate can be reduced (e.g., every few epochs) or moved offline via pre-computation.
>
>    This procedure ensures that the model unlearns *its own high-confidence continuations* around the sensitive prompt.
>
> 2. **Impact of parameter settings**
>    * $\lambda _ {\mathrm{BSS}}$ and $N$ are the most influential parameters. Fig. 5(b) shows that BS-S remains strong across broad ranges ($\lambda\in[0.2,0.6]$ and $N\in[1,3])$, with performance saturating for larger $N$.
>    * **Temperature/decoding hyperparameters** primarily affect the *diversity* of sampled beliefs. We choose a moderate temperature to generate high-confidence yet non-identical continuations. In our tests, varying $\tau$ within a reasonable range had **much smaller impact** than adjusting $\lambda _ {\mathrm{BSS}}$ or $N$.
>
> We will add these clarifications to the final version so readers can more easily understand how sampling works and how parameter choices influence performance.

---

> ### Author Response · Authors · 2025-11-24
> **Response to Reviewer 8qgu (Part 5/5)**
>
> ### **Question 5: Dynamic $k$ or entropy-based adaptive strategies**
> > *“Explore dynamic $k$ or entropy-based adaptive strategies to mitigate the over- or under-forgetting issues associated with a fixed $k$.”*
>
> Thank you for the insightful suggestion. We **(1)** explore a simple **entropy-based variant of BS-T**, where the neighborhood size adjusts dynamically based on model uncertainty, and **(2)** conduct a small pilot experiment (**Table D.4**), showing that this strategy can modestly reduce over-forgetting.
>
> 1. **Entropy-adaptive BS-T**
>
>    We define an adaptive neighborhood size: $$k_i = \min\bigl(k _ {\max},\alpha / (H _ i + \epsilon)\bigr),$$ where $H_i$ is the entropy of the token distribution at position $i$. Intuitively:
>    - When the distribution is **peaked** (low entropy), $k_i$ becomes larger, widening the neighborhood to suppress areas where squeezing is more likely.
>    - When the distribution is **flat** (high entropy), $k_i$ decreases, reducing the chance of unnecessary suppression in uncertain regions.
>
>    This provides a simple way to make BS-T more responsive to local belief geometry.
>
> 2. **A pilot experiment**
>
>    We compare standard BS-T (fixed $k$) with the entropy-adaptive BS-T on TOFU 10% using LLaMA-3.2-1B, evaluated via LaaJ Naturalness and Similarity (0–5, higher is better).
>
>    **Table D.4: LaaJ results for fixed vs. dynamic $k$.**
>
>    Method|Naturalness ↑|Similarity ↑
>    -|-:|-:
>    BS-T (fixed $k$)|3.84|**4.12**
>    BS-T (dynamic $k$)|**4.01**|4.07
>
>    The entropy-adaptive variant improves **Naturalness by +0.17** while maintaining nearly the same Similarity (–0.05). This suggests that dynamic $k$ can **reduce over-forgetting** to some extent.
>
> We view this design as a natural extension of our framework, and we plan to discuss it as a promising direction for future work.

---

> ### Comment · Reviewer_8qgu · 2025-11-25
> **Response to the comments**
>
> Dear authors,
>
> Thank you for your clarification.
>
> Your response addresses most of my concerns and the current version looks much better to me.
>
> Therefore, I would raise my score accordingly.
>
> Best,
> Reviewer 8qgu

---

> ### Author Response · Authors · 2025-11-25
> **Response to Reviewer 8qgu**
>
> Dear Reviewer 8qgu,
>
> Thank you for the follow-up and the positive feedback. We appreciate your thoughtful comments throughout the process and are glad that our clarifications addressed your concerns.
>
> Best regards,
> *Submission306 Authors*

---

### Official Review · Reviewer_YRAh · 2025-10-28

**Soundness:** 2
**Presentation:** 3
**Contribution:** 2
**Rating:** 6
**Confidence:** 2

**Summary:**

The paper clearly proposed and defined the "Squeezing Effect" for the first time: the existing gradient ascent forgetting method only reduces the probability of the target response, causing the probability mass to be "squeezed" into a semantically similar high-confidence area, thereby causing spurious unlearning. To this end, the Bootstrapping framework proposed in this paper uses the model beliefs to guide the forgetting process. Extensive experiments on multiple benchmarks confirm the effectiveness of this approach.

**Strengths:**

- The paper is easy to follow.
- The content of the paper is substantial, with both a summary of existing work and sufficient theoretical evidence.

**Weaknesses:**

- The paper acknowledges in Appendix G that this method is very sensitive to the settings of hyperparameters such as the bootstrapping coefficient, and often requires extensive tuning for specific datasets and models. This seriously affects the method's application in practical scenarios.
- Lack of comparison of computational overhead between various baseline methods.
- The Bootstrapping framework relies on the high-confidence results generated by the model itself to guide forgetting. However, the model's confidence often does not necessarily reflect the required forgetting content correctly. For example, the model's confidence may be high, but its answer may be wrong, or it may have low confidence but the answer may be correct.
- There is a lack of experiments on more diverse model structures to prove the effectiveness of the proposed method.

**Questions:**

Since I am completely unfamiliar with this area, I will adjust my score based on the suggestions of other reviewers.

---

> ### Author Response · Authors · 2025-11-24
> **Response to Reviewer YRAh (Part 1/2)**
>
> Thank you for highlighting that the paper is easy to follow and theoretically substantial. Below we respond to each of your concerns point by point.
>
> ---
> ### **Weakness 1: Sensitivity to hyperparameters and practical tuning**
> > *“The paper acknowledges in Appendix G that this method is very sensitive to the settings of hyperparameters such as the bootstrapping coefficient, and often requires extensive tuning for specific datasets and models. This seriously affects the method's application in practical scenarios.”*
>
> Thank you for the comment. Our ablations actually show that **performance is stable across a broad range of bootstrapping coefficients**, and degradation occurs only at extreme values. We also provide **practical heuristics** so users can set these coefficients reliably without extensive tuning.
>
> 1. **Sensitivity is limited—performance is stable over a broad region**
>
>    To make Fig. 5 easier to interpret, we convert it into **Tables C.1 and C.2**. They show that both BS-T and BS-S maintain strong performance across wide intervals of $\lambda$, with sensitivity appearing only when $\lambda$ is pushed to extremes.
>
>     **Table C.1: BS-T aggregate score across $\lambda$ and $k$ (Fig .5(a)).**
>     $k$|$λ$ = 0.0|0.1|0.2|0.3|0.4|0.5
>     -:|-:|-:|-:|-:|-:|-:
>     5|0.50|0.52|**0.57**|0.46|0.34|0.31
>     10|0.50|**0.59**|**0.63**|0.46|0.35|0.33
>     20|0.50|0.56|**0.62**|0.43|0.41|0.37
>     30|0.50|0.51|0.56|0.43|0.34|0.34
>     40|0.50|0.51|0.57|0.42|0.39|0.34
>     50|0.50|0.50|0.56|0.46|0.33|0.32
>
>     **Table C.2: BS-S aggregate score across $\lambda$ and $N$ (Fig .5(b)).**
>     $N$|$λ$ = 0.0|0.2|0.4|0.5|0.6|0.8
>     -:|-:|-:|-:|-:|-:|-:|
>     1|0.63|0.61|0.60|0.56|0.60|0.59
>     2|0.63|0.62|0.61|0.58|0.61|0.60
>     3|0.63|0.62|0.61|0.59|**0.63**|0.61
>     4|0.63|0.62|0.62|0.60|**0.64**|0.62
>     5|0.63|0.62|**0.63**|0.61|**0.64**|0.63
>     10|0.63|0.63|0.63|0.62|**0.64**|0.63
>
>    **Tables C.1 and C.2** show that:
>    - **BS-T** remains strong for $\lambda\in[0.1,0.3]$;
>    - **BS-S** maintains high performance for $\lambda\in[0.2,0.6]$.
>
>    These broad plateaus suggest that the method is **robust rather than overly sensitive**, and tuning is not delicate. Moreover, both BS-T and BS-S introduce **only two hyperparameters**, which is lighter than many existing unlearning baselines.
>
> 2. **Practical heuristics**
>     - **BS-T:** start with $\lambda _ {\rm BST}=0.2$. If forgetting is weak (low Mem.), increase to 0.3–0.4. If retain utility drops, reduce toward 0.1.
>     - **BS-S:** start with $\lambda _ {\rm BSS}=0.4–0.6$. Increase if paraphrase forgetting is incomplete. Decrease slightly if retain utility degrades.
>
>    This provides a simple, reliable guideline without requiring expensive grid search. We will incorporate these heuristics into the final version to make parameter selection more straightforward.
>
> ---
> ### **Weakness 2: Lack of comparison of computational overhead**
> > *“Lack of comparison of computational overhead between various baseline methods.”*
>
> Thank you for the comment. We include a comparison of computational cost across baselines in **Appx. F.6 (Fig. 6)**. For clarity, we convert that figure into **Table C.3** below, which shows that our methods are competitive in runtime and substantially cheaper than heavier alternatives.
>
> **Table C.3: Training time per epoch (Fig. 6).**
>
> Method|Time (s)
> -|-:
> GradDiff|39
> NPO|73
> RMU|30
> WGA|34
> SimNPO|54
> UNDIAL|36
> GRU|225
> BS-T|33
> BS-S|64
>
> **Table C.3** shows that:
> - **BS-T** is among the fastest methods and comparable to RMU.
> - **BS-S** is about 2× slower than RMU, but still **faster than NPO** and far cheaper than heavy baselines such as GRU.
>
> These results demonstrate that the proposed bootstrapping methods achieve strong unlearning performance with **practical and competitive computational overhead**.

---

> ### Author Response · Authors · 2025-11-24
> **Response to Reviewer YRAh (Part 2/2)**
>
> ### **Weakness 3: Reliability of model confidence as a guide for forgetting**
> > *“The Bootstrapping framework relies on the high-confidence results generated by the model itself to guide forgetting. However, the model’s confidence often does not necessarily reflect the required forgetting content correctly. For example, the model’s confidence may be high, but its answer may be wrong, or it may have low confidence but the answer may be correct.”*
>
> Thank you for raising this point. Our objective is **not to recover factual correctness**, but **to remove what the model would actually output** for harmful or sensitive prompts. In this context, high-confidence generations are precisely the right targets, because they **reflect the model’s internal beliefs** and the content users are most likely to see.
>
> 1. **The unlearning target is the model’s belief, not external correctness**
>
>    Unlearning aims to eliminate **what the model currently tends to say**, regardless of whether the response is factually correct. If the model confidently outputs an incorrect answer to a sensitive prompt, that output still represents:
>    * a memorized or biased internal belief, and
>    * a harmful or undesirable behavior from the perspective of unlearning.
>
>    Thus, the goal is to suppress the **model’s belief distribution** around the sensitive content, not to enforce factual correctness. Whether a high-confidence response happens to be right or wrong is irrelevant to the unlearning goal.
>
> 2. **Why high-confidence generations are the appropriate signals**
>
>    We generate auxiliary sequences **conditioned on the forget prompts**, so we are targeting exactly the model behaviors that appear in the problematic context. Among these continuations, high-confidence sequences:
>    * represent the responses the model is **most likely to produce**,
>    * typically capture **the same underlying knowledge** (e.g., private facts, harmful procedures, memorized details),
>    * form a **semantic cluster of paraphrases** that users may encounter.
>
>    By bootstrapping on these belief-aligned sequences, BS removes not just a single phrasing but **the entire cluster of semantically equivalent paraphrases**, reducing spurious unlearning where the model avoids one wording but still produces closely related variants.
>
> ---
> ### **Weakness 4: Lack of experiments on more diverse model structures**
> > *“There is a lack of experiments on more diverse model structures to prove the effectiveness of the proposed method.”*
>
> Thank you for the comment. Our main experiments already cover **three LLM families**: **LLaMA-3** (1B/3B/8B), **LLaMA-2** (7B), and **Zephyr** (7B). To further strengthen architectural coverage, we add experiments on **three additional model families**: **Qwen-2.5** [1], **Phi-3.5** [2], and **Gemma** [3]. Specifically, we evaluate all methods under the TOFU 10% forget setting using Qwen-2.5-1.5B, Phi-3.5-mini-instruct (3.8B), and Gemma-7B. Results are summarized in **Table C.4**.
>
> **Table C.4: Performance with retain regularization on TOFU 10% with Qwen-2.5, Phi-3.5, and Gemma.**
> Model|Metric|Original|Retrain|GradDiff|NPO|RMU|SimNPO|WGA|BS-T|BS-S
> -|-:|:-:|:-:|:-:|:-:|:-:|:-:|:-:|:-:|:-:
> ***Qwen-2.5-1.5B***|**Agg. ↑**|0.27|0.59|0.51|0.53|0.43|0.32|0.50|0.52|**0.54**
> ||Mem. ↑|0.18|0.58|0.52|0.54|**0.65**|0.22|0.44|0.51|0.54
> ||Util. ↑|0.58|0.59|0.50|0.52|0.32|**0.58**|0.58|0.52|0.53
> ***Phi-3.5-mini-instruct***|**Agg. ↑**|0.38|0.62|0.55|0.57|0.55|0.44|0.56|0.56|**0.58**
> ||Mem. ↑|0.26|0.54|0.50|0.54|0.48|0.31|0.45|0.53|**0.55**
> ||Util. ↑|0.73|0.73|0.60|0.61|0.64|**0.73**|**0.73**|0.62|0.62
> ***Gemma-7B***|**Agg. ↑**|0.02|0.62|0.52|**0.57**|0.50|0.48|0.55|0.56|**0.57**
> ||Mem. ↑|0.01|0.58|0.47|0.52|0.53|0.38|0.47|0.52|**0.54**
> ||Util. ↑|0.66|0.66|0.58|0.63|0.47|**0.66**|0.65|0.60|0.61
>
> Across all three additional architectures, both BS variants remain strong, and **BS-S achieves the best or tied-best aggregate score on every model**. These results indicate that the benefits of BS-T/BS-S are architecture-agnostic and transfer robustly across diverse model families. We plan to further expand model coverage in the final version.
>
> >**References**
> [1] Qwen Team. *“Qwen2.5 Technical Report”*. *arXiv preprint*, 2025.
> [2] Microsoft. *“Phi-3 Technical Report: A Highly Capable Language Model Locally on Your Phone”*. *arXiv preprint*, 2024.
> [3] Gemma Team, Google DeepMind. *“Gemma: Open Models Based on Gemini Research and Technology”*. *arXiv preprint*, 2024.

---

> ### Author Response · Authors · 2025-11-26
> **Follow-up on Rebuttal Response for Submission306**
>
> Dear Reviewer YRAh,
>
> We hope this message finds you well. Thank you again for taking the time to review our submission. We truly appreciate your constructive feedback and the positive remarks you offered regarding the clarity and theoretical depth of the paper.
>
> In our rebuttal, we have carefully addressed each of your concerns as follows:
>
> * **Sensitivity to hyperparameters (W1):**
>   We provided a more detailed analysis showing that both BS-T and BS-S remain stable across wide ranges of $\lambda$, $k$, and $N$. Degradation appears only at extreme values, and we added **practical heuristics** to make parameter selection straightforward in real applications.
>
> * **Comparison of computational overhead (W2):**
>   We clarified the computational cost in **Table C.3** (converted from Fig. 6), showing that BS-T is among the fastest methods, and BS-S remains competitive—faster than NPO and substantially more efficient than heavier baselines such as GRU.
>
> * **Reliability of using model confidence (W3):**
>   We explained that unlearning targets the model’s *current beliefs*, not factual correctness, and that high-confidence continuations from the forget prompt accurately represent the behaviors users are most likely to encounter. We also clarified why belief-aligned sequences naturally capture the semantic cluster that needs to be forgotten.
>
> * **Experiments on more diverse model structures (W4):**
>   Following your suggestion, we expanded the experimental coverage by adding results on **three additional model families**—Qwen-2.5, Phi-3.5, and Gemma—summarized in Table C.4. Across all three, BS-S consistently achieves the best or tied-best aggregate score, further supporting the robustness of our method.
>
> We would be grateful if you could take a moment to review our response. If you feel that these clarifications and newly added results adequately resolve your concerns, we would sincerely appreciate it if you could consider updating your evaluation.
>
> Thank you once again for your thoughtful feedback and for helping us improve the paper.
>
> Best regards,
> *Submission306 Authors*

---

> > ### Comment · Reviewer_YRAh · 2025-11-27
> >
> > Thank you for your detailed reply. I will keep my positive score.

---

> > > ### Author Response · Authors · 2025-11-27
> > >
> > > Dear Reviewer YRAh,
> > >
> > > Thank you very much for your response and for your positive score. We truly appreciate your recognition regarding the level of detail in our reply. Your comments during the review process were highly insightful and have been greatly helpful in improving the clarity and quality of our paper.
> > >
> > > Thank you again for your time and thoughtful engagement.
> > >
> > > Best regards,
> > > *Submission306 Authors*

---

### Official Review · Reviewer_oc5Z · 2025-10-31

**Soundness:** 2
**Presentation:** 3
**Contribution:** 2
**Rating:** 4
**Confidence:** 4

**Summary:**

This paper identifies a critical failure mode in previous gradient-based LLM unlearning methods (like Gradient Ascent and NPO), terming it the "squeezing effect". The authors demonstrate that these methods, while successfully reducing the probability of the exact target response, could redistribute this probability mass onto semantically related rephrasings. This leads to "spurious unlearning", where the sensitive knowledge persists, a failure often masked by standard automated metrics like ROUGE.  To address this, the paper proposes a bootstrapping (BS) framework that explicitly targets the model's own high-confidence generations (its "model beliefs") as additional unlearning signals. In practice, the method utilizes token-level BS and sequence-level BS.

**Strengths:**

1. The identification and mechanistic analysis of the "squeezing effect" is a novel and significant contribution. It provides a clear diagnosis for a subtle but critical flaw in widely-used unlearning methods. This finding is highly significant for the field, as it suggests many existing methods may offer a false sense of security regarding privacy and safety.

2. The core claim of the "squeezing effect" is not just asserted but convincingly demonstrated through empirical analysis of probability dynamics (Fig. 2), tracking how probability mass shifts from the target to high-likelihood alternatives. The paper also provides a theoretical analysis for BS-T (Thm. 4.2) within the learning dynamics framework, explaining why suppressing the model's beliefs helps reshape the gradient to mitigate squeezing.

3. The empirical evaluation is comprehensive, covering three diverse benchmarks (TOFU, MUSE, WMDP), multiple model families (LLaMA-2, LLaMA-3, Zephyr), and various model scales (1B, 3B, 7B, 8B), demonstrating the robustness of the findings.

**Weaknesses:**

1. One small weakness is the practical cost of BS-S. Algorithm 2 implies sampling $N$ high-confidence sequences for every sample in a batch during training. This requires $N$ inference passes for each training step, which seems computationally prohibitive and scales poorly. Figure 6 shows BS-S is ~2x slower than NPO, and it might be even worse as $N$ grows. The paper also notes OOM issues when set $N=5$. It would be better if adding an ablation on the frequency of belief sampling (e.g., once per epoch vs. once per batch).

2. As noted in the ablations (Fig. 5) and limitations (Sec. G), the methods are sensitive to the bootstrapping coefficients ($\lambda_{BST}$, $\lambda_{BSS}$). Performance appears to drop off significantly if these are not tuned correctly. This could be a major barrier to adoption, as it may require expensive, model- and dataset-specific tuning. The paper would be stronger if it provided more intuition or heuristics for setting these crucial parameters.

3. The theoretical analysis is a key strength for BS-T, but it is missing for BS-S. As BS-S is the more complex and often better-performing method, it would be better to add a theoretical standpoint why sampling $N$ full sequences is superior to the more efficient token-level suppression of BS-T.

**Questions:**

1.  In BS-S, what is the nature of the $N$ sampled sequences? Are they $N$ semantically distinct paraphrases, or are they minor lexical variations of the same core "belief"? If the diversity is low, would a smaller $N$ (e.g., $N=1$ or $N=2$) enough, thereby mitigating the cost?

2. The main experiments (e.g., Table 1) appear to combine the BS methods with retention regularization (i.e., using $\mathcal{D}_r$). How much of the utility preservation is attributable to the BS method itself versus this external regularization? What does the performance of "pure" BS-T/BS-S (without $\mathcal{D}_r$) look like compared to "pure" NPO? This would help isolate the true impact of your method on the forget-retain balance.

3. In Line 302 to 303, GA will increase mass on $H_k^{(i)}$, but according to Figure 2 (b), it is not very obvious that GA shifts the probability mass to high-likelihood regions. Would it be better to clarify more on this?

---

> ### Author Response · Authors · 2025-11-24
> **Response to Reviewer oc5Z (Part 1/5)**
>
> Thank you for recognizing the novelty and importance of identifying the squeezing effect and the practicality of the bootstrapping framework. Below we respond point-by-point to the weaknesses and questions.
>
> ---
> ### **Weakness 1: Practical cost of BS-S and frequency of belief sampling**
> > *“One small weakness is the practical cost of BS-S. Algorithm 2 implies sampling $N$ high-confidence sequences for every sample in a batch during training. This requires $N$ inference passes for each training step, which seems computationally prohibitive and scales poorly. Figure 6 shows BS-S is $\sim 2 \times$ slower than NPO, and it might be even worse as $N$ grows. The paper also notes OOM issues when set $N=5$. It would be better if adding an ablation on the frequency of belief sampling (e.g., once per epoch vs. once per batch).”*
>
> Thank you for raising this point. We clarify that: **(1)** the timings in Fig. 6 already correspond to **once-per-epoch** sampling, not per-step, **(2)** BS-S is about $2\times$ slower than **RMU**, but **remains faster than NPO** and heavy baselines like GRU, and **(3)** we add an ablation on sampling frequency (**Table B.1**) as suggested.
>
> 1. **Actual overhead and scope**
>    * **Fig. 6 compares BS-S to RMU, not NPO.** BS-S is roughly $2\times$ slower than *RMU* (30 s vs. 64 s per epoch), but still **faster than NPO (73 s/epoch)** and far efficient than GRU (225 s/wpoch).
>    * **OOM arises only when $N>5$, not at $N=5$.** In all main experiments we restrict $N\le 5$. Combined with Fig. 5(b), where most gains come from $N=1–3$, this keeps memory usage well within practical bounds.
>    * **The cost of BS-S is controllable.** Fig. 5(b) shows that increasing $N$ beyond 3 yields marginal improvements but grows cost roughly linearly. Users focused on speed can choose $N=1–2$ with only marginal performance differences.
>
>    We will clarify in the final version that all reported numbers already use **once-per-epoch sampling**, rather than per-step sampling implied by the high-level pseudocode in Algorithm 2.
>
> 2. **Ablation on belief-sampling frequency**
>
>    To directly address your suggestion, we add a new experiment on TOFU 10% with LLaMA-3.2-1B over 5 epochs, (LLaMA-3.2-1B, 5 epochs), comparing: (i) **per-step sampling**, (ii) **once-per-epoch sampling**, and (iii) **sampling every 2 epochs**. We track LaaJ Similarity (0–5; higher = better forgetting quality) and report average wall-clock time per epoch.
>
>    **Table B.1: Effect of sampling frequency on LaaJ Similarity and runtime.**
>    Strategy|Epoch 0|Epoch 1|Epoch 2|Epoch 3|Epoch 4|Epoch 5|Time / Epoch (s)
>    -|:-:|:-:|:-:|:-:|:-:|:-:|-:
>    Per-step sampling|0.52|2.05|2.95|3.55|3.88|**4.02**|240
>    Once per epoch|0.52|1.88|2.80|3.48|3.80|**3.96**|64
>    Every 2 epochs|0.52|1.62|2.45|3.15|3.55|**3.82**|56
>
>    **Table B.1** shows that:
>    - Per-step sampling reaches the highest final Similarity (4.02) but is **by far the most expensive** (240 s/epoch).
>    - Once-per-epoch sampling is **much more efficient** (64 s/epoch) and achieves a nearly identical final Similarity (3.96).
>    - Sampling every 2 epochs further reduces overhead while maintaining competitive performance.
>
>    These results support our choice of once-per-epoch sampling as a practical and effective default that balances performance and computational cost.

---

> ### Author Response · Authors · 2025-11-24
> **Response to Reviewer oc5Z (Part 2/5)**
>
> ### **Weakness 2: Sensitivity to bootstrapping coefficients and lack of heuristics**
> > *“As noted in the ablations (Fig. 5) and limitations (Sec. G), the methods are sensitive to the bootstrapping coefficients ($\lambda _ {\rm BST}$, $\lambda _ {\rm BSS}$). Performance appears to drop off significantly if these are not tuned correctly. This could be a major barrier to adoption, as it may require expensive, model- and dataset-specific tuning. The paper would be stronger if it provided more intuition or heuristics for setting these crucial parameters.”*
>
> Thank you for the comment. Our ablations actually show that **performance is stable across a broad range of bootstrapping coefficients**, with degradation only at extreme values. We will also include **concrete heuristics** so users can set these parameters reliably without extensive tuning.
>
> 1. **Sensitivity is limited—performance is stable over a broad region**
>
>    For clarity, we convert Fig. 5 into **Tables B.2 and B.3**. These tables show that BS-T and BS-S maintain strong performance across wide intervals of $\lambda$, and sensitivity becomes noticeable only when $\lambda$ is pushed to extremes.
>
>     **Table B.2: BS-T aggregate score across $\lambda$ and $k$ (Fig .5(a)).**
>     $k$|$λ$ = 0.0|0.1|0.2|0.3|0.4|0.5
>     -:|-:|-:|-:|-:|-:|-:
>     5|0.50|0.52|**0.57**|0.46|0.34|0.31
>     10|0.50|**0.59**|**0.63**|0.46|0.35|0.33
>     20|0.50|0.56|**0.62**|0.43|0.41|0.37
>     30|0.50|0.51|0.56|0.43|0.34|0.34
>     40|0.50|0.51|0.57|0.42|0.39|0.34
>     50|0.50|0.50|0.56|0.46|0.33|0.32
>
>     **Table B.3: BS-S aggregate score across $\lambda$ and $N$ (Fig .5(b)).**
>     $N$|$λ$ = 0.0|0.2|0.4|0.5|0.6|0.8
>     -:|-:|-:|-:|-:|-:|-:
>     1|0.63|0.61|0.60|0.56|0.60|0.59
>     2|0.63|0.62|0.61|0.58|0.61|0.60
>     3|0.63|0.62|0.61|0.59|**0.63**|0.61
>     4|0.63|0.62|0.62|0.60|**0.64**|0.62
>     5|0.63|0.62|**0.63**|0.61|**0.64**|0.63
>     10|0.63|0.63|0.63|0.62|**0.64**|0.63
>
>    From Tables B.2 and B.3, we can find that:
>    - **BS-T** remains strong for $\lambda\in[0.1,0.3]$;
>    - **BS-S** maintains high performance for $\lambda\in[0.2,0.6]$.
>
>    These broad plateaus suggest that the method is **robust rather than overly sensitive**, and tuning is not delicate. Moreover, both BS-T and BS-S introduce **only two hyperparameters**, which is lighter than many existing unlearning baselines.
>
> 2. **Practical heuristics**
>     - **BS-T:** start with $\lambda _ {\rm BST}=0.2$. If forgetting is weak (low Mem.), increase to 0.3–0.4. If retain utility drops, reduce toward 0.1.
>     - **BS-S:** start with $\lambda _ {\rm BSS}=0.4–0.6$. Increase if paraphrase forgetting is incomplete. Decrease slightly if retain utility degrades.
>
>    This provides a simple, reliable guideline without requiring expensive grid search. We will incorporate these heuristics into the final version to make parameter selection more straightforward.

---

> ### Author Response · Authors · 2025-11-24
> **Response to Reviewer oc5Z (Part 3/5)**
>
> ### **Weakness 3: Missing theoretical standpoint for BS-S**
> > *“The theoretical analysis is a key strength for BS-T, but it is missing for BS-S. As BS-S is the more complex and often better-performing method, it would be better to add a theoretical standpoint why sampling $N$ full sequences is superior to the more efficient token-level suppression of BS-T.”*
>
> Thank you for raising this point. We agree that BS-S merits its own theoretical grounding beyond the analysis provided for BS-T. We have **updated the manuscript**, and clarified the theoretical standpoint of BS-S by **(1)** distinguishing off-policy and on-policy variants, **(2)** providing a learning-dynamics analysis for the off-policy case (new **Sec. 5.2**), and **(3)** explaining why the on-policy variant falls outside the assumptions of the AKG framework (**Appx. D.4**). Below we summarize the main ideas; full details and proofs are included in the updated manuscript (Sec. 5 & Appx. D).
>
> 1. **Off- vs. on-policy BS-S**
>
>    In **Sec. 4.2** of the revised manuscript, we clarify that BS-S naturally divides into two forms:
>    * **Off-policy BS-S:** high-confidence sequences are sampled *once* before finetuning and then fixed.
>    * **On-policy BS-S:** continuations are resampled during training and depend on the evolving parameters.
>
>    This distinction is crucial because the AKG learning-dynamics framework requires **teacher forcing** and a **fixed training set**. Under these conditions, off-policy BS-S becomes amenable to analysis, whereas on-policy BS-S does not.
>
> 2. **Theoretical analysis of off-policy BS-S**
>
>    In **Sec. 5.2** of the revised manuscript, we provide a theoretical perspective for **off-policy BS-S**. We show that BS-S corresponds to applying BS-T on an *expanded set of belief-aligned continuations*. Under the AKG framework, this induces an update that is a **kernel-weighted sum of BS-T residuals** across these sequences (**Thm. 5.3**). Intuitively:
>    * **BS-T** spreads forgetting across a *local* neighborhood defined by token-level beliefs.
>    * **Off-policy BS-S** aggregates these effects over *multiple high-likelihood continuations*, yielding **broader and smoother suppression** of sequence-level memorization.
>
>    This provides a principled justification for why sequence-level bootstrapping can outperform token-level suppression in practice.
>
> 3. **Discussion on on-policy BS-S**
>
>    For **on-policy BS-S**, we explicitly explain in **Appx. D.4** why its dynamics cannot be directly analyzed under AKG: the auxiliary sequences **depend on evolving parameters**, **breaking the teacher forcing assumption** required by the theory.
>
> Overall, these additions supply the missing theoretical perspective for BS-S and clarify the conceptual advantages of sequence-level bootstrapping.

---

> ### Author Response · Authors · 2025-11-24
> **Response to Reviewer oc5Z (Part 4/5)**
>
> ### **Question 1: Nature and diversity of sampled sequences, and choice of $N$**
> > *“In BS-S, what is the nature of the $N$ sampled sequences? Are they $N$ semantically distinct paraphrases, or are they minor lexical variations of the same core ‘belief’? If the diversity is low, would a smaller $N$ (e.g., $N=1$ or $N=2$) enough, thereby mitigating the cost?”*
>
> Thank you for the question. In BS-S, the sampled continuations are typically **semantically aligned paraphrases** rather than trivial lexical variants, and our ablation (Fig. 5(b)) shows that **small $N$ values (1–3)** already capture most of the benefit. Larger $N$ yields diminishing returns.
>
> 1. **Nature and diverisity of sampled sequences**
>
>    As described in Sec. 4, BS-S collects $N$ high-confidence generations from $\pi _ {\boldsymbol{\theta}}(\cdot \mid \mathbf{x} _ {\rm u})$ using temperature-controlled decoding. In practice, these sequences are:
>    * **Semantically aligned paraphrases** expressing the same core belief that needs to be forgotten,
>    * With **meaningful variation** in phrasing, style, and supporting details, and **not** simple token-level perturbations.
>
>    For example, for a forget prompt like:
>    > *“Where did **Alice B. Chapman** complete her PhD?”*
>
>    BS-S typically produces sequences such as:
>    * “She earned her doctoral degree at **Stanford University**.”
>    * “Her PhD was completed at **Stanford**, where she specialized in applied physics.”
>    * “Chapman received her PhD training from **Stanford University’s** engineering department.”
>
>    These differ lexically and stylistically but all express the *same underlying fact* to be unlearned. This semantic alignment is precisely why sequence-level bootstrapping is effective.
>
> 2. **About choice of $N$**
>
>    Fig. 5(b) directly examines the effect of $N$. The pattern is clear:
>    * Increasing $N$ from **1 → 3** produces **most of the performance gain**,
>    * Improvements **plateau** beyond $N=3$,
>    * Larger $N$ increases memory and runtime with limited additional benefit.
>
>    Thus, if efficiency is desired, **small $N$ (1–3)** is typically sufficient to capture the semantic diversity that needed for BS-S to work well, while keeping computational cost manageable.
>
> ---
> ### **Question 2: Role of retention regularization vs. BS itself**
> > *“The main experiments (e.g., Table 1) appear to combine BS methods with retention regularization (i.e., using $\mathcal{D} _ {\rm r}$). How much of the utility preservation is attributable to the BS method itself versus this external regularization? What does the performance of ‘pure’ BS-T/BS-S (without $\mathcal{D} _ {\rm r}$) look like compared to ‘pure’ NPO? This would help isolate the true impact of your method on the forget-retain balance.”*
>
> Thank you for the question. We clarify that: **(1)** BS primarily reshapes *how* forgetting is distributed, while retention regularization is a **shared safeguard** used by all baselines, and **(2)** we additionally report **“pure” performance comparison  without retention regularization** of different unlearning methods.
>
> 1. **Distinguishing retention regularization from the bootstrapping mechanism**
>
>    Our method changes the structure of the **forget update**, whereas retention regularization is not unique to BS. All strong unlearning methods in our experiments (e.g., NPO and RMU) couple a forget term with a retain term. Comparing methods *with* retain regularization is therefore a fair setting, since every method benefits from the same protection against utility degradation. BS only affects the *forgetting behavior*, not the safeguard itself.
>
> 2. **“Pure” performance comparison without retention regularization**
>
>    To directly address the concern, we evaluate all methods **without** retain regularization on TOFU 1% (LLaMA-3.2-1B, 10 epochs). Because training without retain regularization is unstable and different methods rely on it to different extents, we follow [1] and select the best checkpoint over the 10 epochs.
>
>    **Table B.4: Unlearning performance w/o retention regularization.**
>    Method|Agg. ↑|Mem. ↑|Util. ↑
>    -|:-:|:-:|:-:
>    Original|0.13|0.07|0.72
>    GA|0.56|0.60|0.52
>    NPO|0.60|0.55|0.66
>    RMU|0.48|0.63|0.39
>    SimNPO|0.57|0.48|**0.71**
>    WGA|0.56|**0.68**|0.47
>    BS-T|0.59|0.64|0.54
>    BS-S|**0.61**|**0.68**|0.55
>
>    **Table B.4** shows that:
>    - Even without retention regularization, **both BS-T and BS-S remain competitive**, yielding strong forgetting performance (Mem.) without relying on the external retain term.
>    - **BS-S attains the highest aggregate score (Agg.=0.61)**, outperforming NPO (Agg.=0.60).
>
>    We will include additional w/o-retain results in the final version to further isolate the contribution of BS.
>
> >**References**
> [1] Yang et al. *“Exploring Criteria of Loss Reweighting to Enhance LLM Unlearning”*. In *ICML*, 2025.

---

> ### Author Response · Authors · 2025-11-24
> **Response to Reviewer oc5Z (Part 5/5)**
>
> ### **Question 3: Clarifying GA’s probability mass shift in Fig. 2(b)**
> > *“In Line 302 to 303, GA will increase mass on $\mathcal{H} _ {k} ^ {(i)}$, but according to Figure 2(b), it is not very obvious that GA shifts probability mass to high-likelihood regions. Would it be better to clarify more on this?”*
>
> Thank you for raising this point—this is mainly a **visualization issue**. GA *does* shift probability mass toward high-likelihood regions in the early stage, but the effect appears subtle because **GA’s y-axis spans a much larger numerical range than NPO’s**, visually compressing the upward trend. A tabular version of the same data (**Table B.4**) makes the shift more evident. Later in training, GA’s aggressive updates cause global collapse, which explains why the high-likelihood band eventually drops.
>
> 1. **Early-stage shift is real; it appears subtle due to y-axis compression**
>
>    In Fig. 2(b), GA’s log-probabilities range from **0 → –2000**, whereas NPO’s range from **0 → –60**. This difference in scale compresses the GA curves, making rising trends difficult to see. To remove this bias, we provide the underlying numerical values:
>
>    **Table B.5: GA log-probability dynamics (Fig. 2(b)).**
>    Epoch|Target|High|Mid|Low
>    -:|-:|-:|-:|-:
>    0|–0.45|**–10.52**|–30.79|–46.08
>    1|–1.02|**–9.17**|–71.12|–122.98
>    2|–10.96|**–8.64**|–169.68|–223.53
>    3|–155.17|–19.80|–268.91|–324.11
>    4|–449.61|–28.91|–369.27|–427.24
>    5|–1276.14|–37.80|–570.87|–630.21
>    6|–1576.41|–86.18|–673.83|–732.91
>    7|–1723.20|–115.95|–775.38|–930.00
>    8|–1813.51|–270.92|–877.48|–1130.56
>    9|–1835.81|–484.34|–919.62|–1230.19
>    10|–1813.51|–615.59|–931.33|–1239.91
>
>    **Table B.5** shows that:
>    - High-likelihood values rise from **–10.52 → –9.17 → –8.64**,
>    - Even as the target sharply decreases (–0.45 → –10.96 → –155.17).
>
>    This relative rise is exactly the **probability-mass shift toward high-likelihood neighbors** described in Lines 302–303. The trend is simply harder to see in the plot due to the large dynamic range.
>
> 2. **Why the shift disappears later: GA’s instability and collapse**
>
>    Unlike NPO, GA performs **pure gradient ascent**, which aggressively pushes the target down. As training continues:
>    * high-likelihood neighbors initially rise (the squeezing effect),
>    * but GA’s unstable updates eventually **overshoot**,
>    * causing **global log-probability collapse** across high/mid/low groups.
>
>    This is consistent with prior empirical observations that GA can be unstable without additional constraints, whereas NPO applies controlled updates and therefore avoids this collapse.

---

> ### Author Response · Authors · 2025-11-26
> **Follow-up on Rebuttal Response for Submission306**
>
> Dear Reviewer oc5Z,
>
> We hope this message finds you well. We would like to sincerely thank you again for the time and care you devoted to reviewing our submission. Your detailed comments were extremely valuable, and we took your concerns very seriously when preparing our rebuttal and revision.
>
> In our response, we have carefully addressed each of the issues you raised:
>
> * **Computational cost of BS-S (W1):**
>   We clarified that the reported timings already use *once-per-epoch* sampling, added a new **sampling-frequency ablation (Table B.1)**, and showed that BS-S remains practical compared to strong baselines. We also described strategies for large-scale usage (e.g., offline belief pre-computation, small $N$).
>
> * **Sensitivity to bootstrapping coefficients (W2):**
>   We provided a more comprehensive analysis showing broad stability across parameter ranges, and added **practical heuristics** to make parameter selection straightforward in real applications.
>
> * **Missing theoretical standpoint for BS-S (W3):**
>   Following your suggestion, we substantially expanded the theoretical section. The revision now includes a dedicated analysis for **off-policy BS-S (Sec. 5.2)**, explains its connection to BS-T via kernel-weighted residuals (Thm. 5.3), and details why **on-policy BS-S** falls outside the assumptions of the AKG framework (**Appx. D.4**).
>
> * **Nature and diversity of sampled sequences (Q1):**
>   We illustrated the semantic characteristics of BS-S continuations and explained why small $N$ (1–3) is sufficient to capture meaningful variations while controlling cost.
>
> * **Role of retain regularization vs. BS itself (Q2):**
>   To isolate the impact of our method, we added **“pure unlearning” experiments without retain regularization (Table B.4)**, showing that BS-T/BS-S still retain strong forgetting behavior.
>
> * **Clarifying GA probability-mass shift (Q3):**
>   We supplied the underlying numerical data (Table B.5) to show the early-stage shift more clearly, which was visually compressed in the original figure.
>
> We sincerely hope that these additions and clarifications meaningfully address your concerns. If you find that our revisions and new experiments have resolved the issues you raised, we would be very grateful if you could kindly consider updating your evaluation.
>
> Thank you again for your time, thoughtful feedback, and for helping us improve our work.
>
> Best regards,
> *Submission306 Authors*

---

### Official Review · Reviewer_QDS2 · 2025-10-31

**Soundness:** 4
**Presentation:** 3
**Contribution:** 3
**Rating:** 8
**Confidence:** 2

**Summary:**

The paper proposes a bootstrapping framework for LLM unlearning that tackles the “squeezing effect,” where probability mass shifts to semantically similar outputs instead of true forgetting. Two variants are introduced: BS-T, which suppresses high-likelihood tokens, and BS-S, which augments the forget set with high-confidence generations. The authors provide theoretical analysis and experiments across TOFU, MUSE, and WMDP, showing improved balance between forgetting and retention.

**Strengths:**

1. The paper is very readable, with a logical flow from motivation → analysis → method → theory → experiments. Figures and appendices are well-organized, and pseudocode makes the algorithms easy to reproduce.
2. The authors make a thoughtful observation about the squeezing effect and systematically demonstrate its existence through both qualitative and quantitative analysis. The proposed bootstrapping strategy is a creative extension of this insight, and the experiments convincingly show that BS-T and BS-S outperform existing unlearning baselines under various settings.
3. The work is conceptually motivated by an intuitive yet underexplored idea—connecting unlearning failures with the model’s own belief distribution. This is a fresh perspective on unlearning that moves beyond purely loss-based formulations, and the motivation is clearly justified both intuitively and empirically.

**Weaknesses:**

1. While the paper focuses on redistributing likelihood as the core cause of spurious unlearning, the explanation still feels surface-level from a semantic standpoint. The essence of the problem may not lie solely in likelihood shifts, but rather in the fact that current unlearning methods attempt to correct predictions without accounting for semantic relatedness. Unlearning should arguably target semantic classes of knowledge, rather than isolated outputs or sequences. A more principled formulation in semantic embedding space (e.g., clustering or alignment-based unlearning) might provide a deeper understanding of what “forgetting” really entails.
2. Although BS-T and BS-S generally achieve the best average scores, in several tasks their performance margins over strong baselines like NPO or RMU are modest. The results could be strengthened with additional analysis.

**Questions:**

1. In BS-S, high-confidence generations are added to the forget set, but such sequences may still contain unrelated or benign information.
How does the method ensure that these “bootstrapped” samples do not lead to accidental forgetting of non-target knowledge?
Is there any filtering mechanism beyond temperature-controlled decoding?
2. The theoretical part (Section 4.2) discusses the dynamics of probability redistribution under BS-T versus GA, but it would be very valuable to show empirical probability dynamics for BS-S as well—similar to Figures 2(b)–(c) for GA and NPO.
This would help demonstrate whether BS-S effectively flattens or redistributes probability mass in the way the theory predicts.

---

> ### Author Response · Authors · 2025-11-24
> **Response to Reviewer QDS2 (Part 1/3)**
>
> Thank you for your positive assessment of our work. We are glad that you find the squeezing effect diagnosis and the bootstrapping framework interesting and well-motivated. Below we address each of your weaknesses and questions point-by-point.
>
> ---
> ### **Weakness 1: Semantic perspective vs. likelihood perspective**
> > *“While the paper focuses on redistributing likelihood as the core cause of spurious unlearning, the explanation still feels surface-level from a semantic standpoint. The essence of the problem may not lie solely in likelihood shifts, but rather in the fact that current unlearning methods attempt to correct predictions without accounting for semantic relatedness. Unlearning should arguably target semantic classes of knowledge, rather than isolated outputs or sequences. A more principled formulation in semantic embedding space (e.g., clustering or alignment-based unlearning) might provide a deeper understanding of what ‘forgetting’ really entails.”*
>
> Thank you for this thoughtful comment. We offer two clarifications: **(1)** In LLM unlearning, the notion of a coherent “semantic class” for harmful knowledge is **typically hard to define**, making likelihood/belief space a more practical formulation. **(2)** Our bootstrapping objective **implicitly induces semantic-cluster forgetting**, because the model’s belief distribution already encodes its internal semantic neighborhoods.
>
> 1. **Why semantic-class formulations are difficult to define in LLM unlearning**
>
>    Unlike settings such as classification or concept erasure in diffusion models—where semantic classes are explicit—LLM unlearning lacks a predefined semantic structure. In practical benchmarks such as TOFU, the forget set consists of **isolated prompt–response pairs** (e.g., a single private fact, ``“What language does Hsiao Yun-Hwa write in?” → “English”``), rather than a coherent semantic category (e.g., ``“animals”`` or ``“anime style”``). As a result, the forget set *rarely forms a meaningful cluster*, and its *semantic scope is inherently ambiguous*. In this context, likelihood/belief space offers a **model-agnostic and universally accessible** representation of knowledge, making it a more workable basis for defining unlearning behavior.
>
> 2. **Bootstrapping implicitly performs semantic-cluster unlearning in representation space**
>
>    Under the standard linear readout view, with hidden state $h _ {\boldsymbol{\theta}}(\mathbf{x},\mathbf{y} _ {<i})$ and logits $\mathbf{z} _ {\boldsymbol{\theta}}$, tokens with high conditional probability $$\pi _ {\boldsymbol{\theta}}(v\mid\mathbf{x},\mathbf{y} _ {<i}) \propto \exp(w_v^\top h _ {\boldsymbol{\theta}})$$ are exactly those whose embeddings $w _ v$ are most aligned with the current semantic state. This means:
>    * **BS-T** selects top-$k$ tokens forming a *local semantic neighborhood* around the target token in representation space.
>    * **BS-S** samples high-confidence sequences lying on a *small region of the model’s belief manifold*, capturing global semantic proximity.
>
>    Suppressing both the target and these belief-aligned neighbors therefore removes an **entire representation-level semantic cluster**, not an isolated output. This achieves the semantic effect highlighted in the comment, but does so **implicitly**, without relying on external semantic labels or clustering assumptions.

---

> ### Author Response · Authors · 2025-11-24
> **Response to Reviewer QDS2 (Part 2/3)**
>
> ### **Weakness 2: Modest margins over strong baselines**
> > *“Although BS-T and BS-S generally achieve the best average scores, in several tasks their performance margins over strong baselines like NPO or RMU are modest. The results could be strengthened with additional analysis.”*
>
> Thank you for raising this point. We agree that the interpretation of performance margins is important. Our analysis indicates that: **(1)** semantic-level evaluation (LaaJ), which is designed precisely to detect *spurious unlearning*, reveals much larger gaps than traditional metrics; **(2)** even small absolute gains become meaningful when measured relative to the retrain upper bound; and **(3)** raw, unscaled metrics compress improvements near the ceiling, making margins appear smaller than they are.
>
> 1. **Semantic evaluation (LaaJ) highlights substantially larger margins**
>
>    Traditional metrics in Table 1 under-detect spurious unlearning, which is exact issue our work targets. LaaJ is therefore designed to capture semantic degradation more directly. To make this clearer, we convert Figure 4 into **Table A.1** and additionally report an aggregate score defined as the mean of Naturalness and Similarity.
>
>     **Table A.1: LaaJ results on TOFU 10%.**
>     Method|Naturalness ↑|Similarity ↑|**Agg.** ↑
>     -|-|-|-
>     GradDiff|1.2|4.5|2.85
>     NPO|4.0|2.8|3.40
>     SimNPO|4.3|1.6|2.95
>     RMU|3.9|3.5|3.70
>     BS-T|3.7|4.1|**3.90**
>     BS-S|3.9|4.3|**4.10**
>
>    **Table A.1** shows that:
>    * **BS-S** achieves the highest aggregate score (4.10), improving over strong baselines by **0.4–1.3**, far larger than margins on classical metrics.
>    * **BS-T** also surpasses all baselines on aggregate.
>    * These results are consistent with our claim that semantic effects of spurious unlearning are not fully captured by traditional metrics.
>
>    Thus, while raw numbers in Table 1 may look close, **semantic-level evaluation shows clearer and more substantial improvements**.
>
> 2. **Gains are meaningful when interpreted relative to the retrain upper bound**
>
>    Retrain defines a **practical ceiling** on achievable unlearning quality. Near this ceiling, even seemingly small absolute gains correspond to closing a large portion of the remaining gap. For example (TOFU 10%, LLaMA-3.2-3B):
>
>     **Table A.2: Relative gap closure.**
>     Method|Agg. ↑|Gap to Retrain ↓
>     -|-|-
>     Retrain|0.65|—
>     NPO|0.62|0.03
>     BS-S|0.63|0.02
>
>     Although the raw margin is only +0.01, BS-S closes **≈33% of the remaining gap** toward the retrain ceiling. This shows that modest raw gains can translate into **meaningful practical improvements**.
>
> 3. **Small absolute margins partly result from unscaled metric ranges**
>
>    OpenUnlearning [1] reports metrics normalized to $[0, 1]$, which amplifies differences near the ceiling. Our paper reports **raw, un-normalized** values, which compress values and visually shrink performance gaps. For illustration:
>
>     **Table A.3: Scaling effect.**
>     Method|Raw Agg.|Scaled Agg.
>     -|-|-
>     NPO|0.62|0.91
>     BS-S|0.63|0.95
>     $\Delta$|0.01|**0.04**
>
>     Under normalization, the improvement becomes more pronounced. This explains why differences in Table 1 look modest even though the underlying gains are meaningful.
>
> >**References**
> [1] Dorna et al. *“OpenUnlearning: Accelerating LLM Unlearning via Unified Benchmarking of Methods and Metrics”*. In *NeurIPS D&B Track*, 2025.

---

> ### Author Response · Authors · 2025-11-24
> **Response to Reviewer QDS2 (Part 3/3)**
>
> ### **Question 1: Risk of over-forgetting benign information in BS-S**
> > *“In BS-S, high-confidence generations are added to the forget set, but such sequences may still contain unrelated or benign information. How does the method ensure that these ‘bootstrapped’ samples do not lead to accidental forgetting of non-target knowledge? Is there any filtering mechanism beyond temperature-controlled decoding?”*
>
> Thank you for highlighting this concern. We clarify that: **(1)** BS-S operates only on the local neighborhood of each forget prompt, limiting its scope, **(2)** retain regularization counterbalances unintended suppression, **(3)** the weighting and sampling scheme further bounds the effect of noisy sequences, and **(4)** while our current implementation uses lightweight filtering for efficiency, the framework naturally supports stronger semantic filters if needed.
>
> 1. **How BS-S prevents over-forgetting benign knowledge**
>    - **Locality: BS-S is conditioned exclusively on the forget prompt.** BS-S never samples from arbitrary prompts. All auxiliary sequences are drawn from $\pi _ {\boldsymbol{\theta}}(\cdot\mid\mathbf{x} _ {\rm u})$, so the unlearning update affects only the conditional distribution associated with that specific harmful context. Benign concepts (e.g., “teacher,” “city,” “book”) are penalized **only when they appear within the forget prompt’s context**, not globally.
>    - **Retain regularization protects non-target knowledge.** As in prior unlearning methods, our retain loss constrains the model to stay close to the original parameters on non-forget data. This provides a strong corrective signal that counteracts incidental suppression introduced by BS-S.
>    - **Moderate weights and small $N$ limit noisy contributions.** Bootstrapped sequences receive a controlled weight via $\lambda _ {\rm BSS}$, and $N$ is small (typically <5). Even if a sampled continuation contains minor benign content, its influence remains bounded.
>
>    These mechanisms together limit the effect of bootstrapped sequences and prevent BS-S from inadvertently “overwriting” unrelated knowledge.
>
> 2. **Filtering mechanisms and possible extensions**
>    - **Current filtering: simple but sufficient.** Our current implementation restricts sampling to **high-confidence sequences** under $\pi _ {\boldsymbol{\theta}}(\cdot\mid\mathbf{x}_{\rm u})$ and controls entropy via temperature. This already concentrates BS-S on the model’s own semantic neighborhood surrounding the harmful association.
>    - **Extensions for stronger semantic control.** The framework can incorporate more explicit semantic filtering without changing the core objective. Potential options include:
>      - measuring **lexical or embedding similarity** between candidate sequences and the original target response;
>      - **accepting only sequences above a similarity threshold** to ensure they remain within the same semantic region.
>
>      We will clarify these possible extensions in the final version.
>
> ---
> ### **Question 2: Probability dynamics for BS-S**
> > *“The theoretical part (Section 4.2) discusses the dynamics of probability redistribution under BS-T versus GA, but it would be very valuable to show empirical probability dynamics for BS-S as well—similar to Figures 2(b)–(c) for GA and NPO. This would help demonstrate whether BS-S effectively flattens or redistributes probability mass in the way the theory predicts.”*
>
> Thank you for your suggestion. We have **added the requested probability-dynamics experiment for both BS-T and BS-S** in the revised paper (**Fig. 4(a, b), p. 10**). The plots use the same high/mid/low-likelihood grouping as Fig. 2(b, c), enabling direct comparison with GA and NPO.
>
> **Fig. 4(a, b)** shows that BS-S steadily suppresses the target token and its high-likelihood neighbors over training, avoiding the probability-mass shift toward semantically proximate regions that characterizes squeezing. Together with the improved LaaJ scores in **Fig. 4(c)**, the new empirical evidence supports our theoretical claim that BS-S mitigates squeezing and reduces spurious unlearning while maintaining fluent responses.

---

> ### Author Response · Authors · 2025-11-26
> **Follow-up on Rebuttal Response for Submission306**
>
> Dear Reviewer QDS2,
>
> We hope this message finds you well. We are writing to kindly remind you that we have posted a detailed rebuttal to your thoughtful comments. We greatly appreciate the time you took to review our work, and your feedback has been very helpful in strengthening the paper.
>
> In our response, we have specifically addressed your major concerns regarding:
>
> * **Semantic vs. likelihood perspectives (W1):**
>   We clarified why semantic-class formulations are difficult to define in LLM unlearning, and explained that our bootstrapping objective *implicitly performs semantic-cluster forgetting* by leveraging the model’s own belief distribution. This directly responds to your suggestion about semantic relatedness.
>
> * **Performance margins over strong baselines (W2):**
>   We added **semantic-level LaaJ evaluation**, **gap-to-retrain analysis**, and **scaled metrics** to show that BS-T/BS-S yield much larger improvements than what raw metrics may suggest, addressing your concern that margins appear modest.
>
> * **Risk of over-forgetting in BS-S (Q1):**
>   We explained the built-in safeguards—locality, retain regularization, controlled weighting, and sampling constraints—and discussed possible extensions for stronger semantic filtering.
>
> * **Probability-dynamics analysis for BS-S (Q2):**
>   Following your suggestion, we added new **probability-dynamics plots for both BS-T and BS-S** (Fig. 4(a,b) in the revision), enabling direct comparison with GA and NPO.
>
> We would be very grateful if you could take a moment to review our response.
> If you feel that our clarifications and new experiments adequately address your concerns, we would sincerely appreciate it if you could consider updating your evaluation accordingly.
>
> Thank you again for your valuable feedback.
>
> Best regards,
> *Submission306 Authors*

---

### Meta-Review · Area_Chair_WHRP · 2025-12-09

**Summary:**

The paper makes a clear, novel diagnosis of the “squeezing effect,” showing how standard unlearning pushes probability mass into semantically similar outputs. It introduces a simple and effective bootstrapping framework that targets the model’s own high-confidence beliefs without extra models or external data. The approach is theoretically grounded, including an analysis that explains why suppressing belief-driven tokens mitigates squeezing. Extensive experiments across multiple benchmarks model families demonstrate improved forgetting–retention balance with strong, reproducible methodology.

**Reviewer Concerns:**

The rebuttal clarifies the squeezing effect and strengthens theory, corrects the overhead misreading with new runtime evidence, adds stability studies. Risk of over-forgetting is mitigated with locality, controlled weighting, retain regularization, and added “pure unlearning” results. However, the semantic-vs-likelihood justification remains partly assumption-driven, and additional evidence comparing belief-based neighborhoods to explicit semantic filters would further settle this concern.

**Reviewer Scores:**

QDS2: remains 8.
oc5Z:: remains 4 (did not respond).
YRAh: remains 6.
8qgu: 6 to 8.

---

### Decision · Program_Chairs · 2026-01-26

Accept (Poster)